# At Which Area Level Does COVID-19 Infection Matter Most for an Individual’s Self-Rated Health? A Multilevel Fixed-Effects Model Analysis in Japan

**DOI:** 10.3390/ijerph19158918

**Published:** 2022-07-22

**Authors:** Takashi Oshio, Hiromi Kimura, Toshimi Nishizaki, Susumu Kuwahara

**Affiliations:** 1Institute of Economic Research, Hitotsubashi University, 2-1 Naka, Kunitachi, Tokyo 186-8603, Japan; 2Survey Research Center, 3-13-5 Nihonbashi, Chuoku, Tokyo 103-0027, Japan; kimura_h@surece.co.jp; 3Japan Ministry of Finance, 3-1-1 Kasumigasei, Chiyoda-ku, Tokyo 100-8914, Japan; toshimi.nishizaki@mof.go.jp; 4Japan Cabinet Office, 1-6-1 Nagatacho, Chiyoda-ku, Tokyo 100-8914, Japan; susumu.kuwahara.y6h@cao.go.jp

**Keywords:** COVID-19, prefecture, regional bloc, self-rated health, state of emergency

## Abstract

Several studies have reported the adverse impacts of the COVID-19 pandemic on health outcomes. However, little is known about which area of COVID-19 infection matters most for an individual’s subjective health outcomes. We addressed this issue in the present study. We used the longitudinal data of 2260 individuals obtained from a two-wave internet-based nationwide survey conducted in Japan. We estimated the multilevel regression models, which controlled for fixed effects at the individual and prefecture levels, to explain an individual’s self-rated health (SRH) based on the reported number of new COVID-19 infection cases at different area levels: prefecture, group of neighboring prefectures, and regional bloc. We found that SRH was highly associated with the average and maximum number of new infection cases among neighboring prefectures or in the regional bloc, but not with those at the prefecture level, if used jointly as explanatory variables. The results suggest that inter-prefectural coordination is needed not only to contain COVID-19 but also to reduce its adverse impact on the subjective health outcomes of residents.

## 1. Introduction

Several studies have provided evidence on the adverse impact of the COVID-19 pandemic on a wide range of health outcomes and health behaviors [1,2,3,4,5]. Even if they are not actually infected with the coronavirus, individuals tend to assess their health negatively due to various pandemic-related factors, including enhanced uncertainties with employment and income conditions [6,7,8,9,10,11] and higher risks of social isolation caused by restricted mobility due to the implementation of lockdowns and social distancing policies [11,12,13,14,15,16,17,18,19,20].

Concerns about the seriousness of the pandemic and its potential impact on health are likely to be affected by the number of new COVID-19 cases, which have been reported in the mass media every day, as well as the declaration of a state of pandemic-related emergency. Hence, we can reasonably predict that individuals living in areas with higher incidence of COVID-19 may feel more stressed and, accordingly, assess their health more negatively than others. In this context, the number of new cases is likely to represent the degree of perceived COVID-19 threat, which has harmed the individual’s mental health as reported in previous studies [21,22,23].

However, little is known about the area in which the reported COVID-19 cases matter most for an individual’s subjective health assessment. Despite living in an area with a low number of COVID-19 cases, individuals may become concerned about the increasing number of cases in neighboring areas or in the regional bloc to which the concerned area belongs, because individuals across the border are not fully immune from the infection. Accordingly, an individual’s subjective assessment of health is affected by the number of infection cases in an area that is wider than where they live. In the same vein, the state of emergency declared in one of the neighboring areas or within the regional bloc may affect the individual’s subjective health assessment, even if they live in an area with no such declaration.

We can also suspect that the comparisons with other areas—especially neighboring ones—may affect an individual’s subjective health assessment, as suggested by the “relative income hypothesis” about the determinants of subjective well-being [24,25,26,27]. This hypothesis argues that subjective well-being, such as life satisfaction and happiness, is affected not only by one’s own income but also by comparisons with others—especially those who have similar socioeconomic attributes. In the context of the COVID-19 pandemic, the underperformance of the area where one lives—that is, the higher numbers of infection cases compared with those in the neighboring areas or regional bloc—may have an additional and independent impact on an individual’s subjective health assessment.

Keeping these possibilities in mind, we evaluated the association between area-level COVID-19 and individual-level self-rated health (SRH) in Japan. The prefecture was regarded as the basic unit of area because the number of infection cases has been reported for each prefecture and the state of pandemic-related emergency was declared at the prefecture level. At a wider area level, we focused on a group of neighboring prefectures that geographically share borders with a concerned prefecture. We also considered the regional bloc, which is conventionally defined. Japan has 47 prefectures, and they are usually divided into seven regional blocs, which consist of prefectures closely located and related to each other in terms of socioeconomic and institutional activities.

We examined the association between an individual’s SRH and the reported numbers of new infection cases at three different area levels: prefecture, neighboring prefectures, and regional bloc. Using two-wave longitudinal data obtained from a nationwide internet survey, analysis was performed after controlling for the fixed (time-invariant) attributes at the individual and prefecture levels [27]. This multilevel fixed-effects regression analysis is expected to clarify the association between individual-level subjective health assessment and area-level infection compared with a cross-sectional analysis.

We tentatively hypothesized that an individual’s SRH is associated with infection cases not only at the prefecture level but also at wider area levels, given the plausibly high risks of infection spread across prefecture borders, although the relative importance of each area level relies entirely on the results of empirical analysis. If this hypothesis is validated, we can argue that inter-prefectural coordination is required not only to contain the infection, but also to reduce its adverse impact on subjective health outcomes.

## 2. Materials and Methods

### 2.1. Study Sample

This study used the data obtained from a population-based, nationwide internet survey conducted in early March 2021 (Wave 1) and in October/November 2021 (Wave 2). The registrants of an internet survey company were included in this study. Approximately three-quarters of the registrants were equally distributed between each prefecture, between men and women, and among the five age groups (15–24, 25–34, 35–44, 45–59, and >60 years). The remaining quarter of the registrants were distributed to each gender-age group in each prefecture in proportion to each prefecture’s actual population size. Accordingly, the sample was not fully representative of the Japanese population. Before the survey was conducted, the questionnaire was pretested to explore potential issues in interpreting the questions.

A Wave 1 survey was conducted between 3 and 11 March 2021, when the third wave of COVID-19 infections occurred in Japan. Four prefectures in the Tokyo metropolitan area (Tokyo, Chiba, Saitama, and Kanagawa) were in a state of emergency and, just after, six prefectures (Aichi, Gifu, Osaka, Hyogo, Kyoto, and Fukuoka) lifted the state of emergency on 28 February 2021. The questionnaires were sent to approximately 4200 registrants, and data were collected from 2311 individuals. A Wave 2 survey was conducted between 28 October and 8 November 2021, a month after all prefectures had lifted their state of emergency on September 30. The questionnaires (see the Appendix A) were sent to those who participated in the survey in Wave 1, which collected data from 2260 individuals who participated in both surveys. The balanced two-wave data were used in the statistical analysis.

### 2.2. Measures

#### 2.2.1. Area Units

Three area levels were considered in this study: prefecture, group of neighboring prefectures, and regional bloc. The latter two overlapped. Each prefecture consists of a group of neighboring prefectures and the prefectures with which it shares its borders. Eight regional blocs, which consist of seven conventionally defined regional blocs (Hokkaido, Totoku, Kanto, Chubu, Kinki, Chugoku/Shikoku, and Kyushu), and Okinawa, which is located far from the Kyushu bloc, were also considered. For Hokkaido and Okinawa, each of them only consists of the neighboring prefectures and regional blocs.

#### 2.2.2. Self-Rated Health

The study focused on SRH as a subjective health outcome. SRH is frequently used in demographic and population health surveys to capture respondents’ self-reported general health including mortality [28,29]. The merit of this self-reported indicator is explained by its ease of use (a single question) and its validity and reliability [28,30,31,32]. With regard to SRH, the survey asked participants to answer the question “How do you feel about your health condition?” and they responded by selecting *good*, *somewhat good*, *average*, *somewhat poor*, or *poor*. A five-point scale was constructed, with scores ranging from 1 (*good*) to 5 (*poor*); a higher score indicates poor SRH. A binary variable for poor SRH was also constructed by allocating 1 to those who answered *poor* and *somewhat poor*, and zero to others, considering that the score is generally skewed toward better health.

#### 2.2.3. COVID-19 Infection Cases

The key independent variables were the monthly total number of new COVID-19 cases in February 2022 and October for Waves 1 and 2, respectively. Their daily numbers were compiled and released by the Ministry of Labour, Health and Welfare (MLHW) every day, and widely reported by the mass media. In addition to the new cases reported in each prefecture, the (unweighted) average and maximum cases for each group of neighboring prefectures and regional blocs were computed.

#### 2.2.4. Covariates

Educational attainment, sex, age, marital status, occupational attainment, and household income were considered as individual-level covariates. As for age, the participants were categorized into those in their 20s or below, 30s, 40s, 50s, and 60s or above. Educational attainment was categorized into junior high school, high school, junior college, and college or higher. Occupational status was categorized into regularly employed, non-regularly employed (part-time or temporary), self-employed, out-of-labor force, unemployed, and students. Binary variables were constructed for each quartile of household income. Sex was mechanically dropped from the regression because it was fixed. In addition, binary variables for the experiences of (i) COVID-19 infection and (ii) other serious disease or injury over the year prior to the survey time were included in the regressions.

### 2.3. Analytic Strategy

The regression models were estimated, which linearly explained the SRH score based on (i) the new COVID-19 infection cases in the prefecture where the participant was living and (ii) the average or maximum cases in a group of neighboring prefectures or a regional bloc, along with the individual-level covariates. This regression analysis was adjusted for fixed effects at the individual and prefecture levels. This study also examined how SRH was affected by the declaration of the state of emergency at different area levels, using a binary variable for the declaration in a prefecture where the participant was living and another binary variable for the declaration in at least one prefecture among the neighboring prefectures or in a regional bloc. A similar regression analysis was performed by replacing the SRH score with a binary variable for poor SRH.

In this regression analysis, inverse probability weighting was applied to mitigate the attrition bias [33], although only 2.2% of the original participants were dropped from the survey in Wave 2. Specifically, the probit model was initially estimated to predict the probability that the participants will stay in the survey until Wave 2 using their attributes observed in Wave 1. The inverse of the predicted probability was used as the weight in the regression analysis.

By setting the effect size = 0.35, *α* = 0.05, the power (1-*β*) = 0.8, and the number of predictors = 2324 (which reflected 19 independent variables, 2260 individual fixed effects, and 47 prefectural fixed effects), the sample size was estimated to be 2462, which was well below the 4520 observations used in our regression analyses.

## 3. Results

Table 1 summarizes the key features of the participants evaluated in Wave 1. The proportions of participants were lower in the Kanto and Kinki blocs and higher in other areas (except for Hokkaido) compared with the actual population reported in 2021. This is because three-quarters of the registrants from whom the study participants were selected were distributed equally between each prefecture, resulting in the lower area of regional blocs having larger populations.

Figure 1 illustrates the distribution of new COVID-19 cases by prefecture in February 2022 and October 2022. In February 2022, the number of cases was highest in the Tokyo metropolitan area, followed by the Kinki area. In October 2022, the number of cases was generally lower, whereas the highest number of cases was reported in Osaka. At both times, some prefectures with low infection rates were located close to prefectures with high infection rates.

Table 2 summarizes the key statistics of the SRH variables and their pairwise correlations with new infection cases at different area levels. Somewhat surprisingly, neither the SRH score nor the probability of poor SRH was correlated with the cases at the prefecture level. Instead, the SRH score was closely related to the average and maximum cases in a group of neighboring prefectures or the regional bloc, and poor SRH was related only to the average and maximum cases in the regional bloc.

This table also presents the correlation between SRH measures and the declaration of a state of emergency. In line with the number of new infection cases, the declaration among neighboring prefectures or in the regional bloc affected the SRH scores compared with the declaration at the prefecture level, and only the declaration in the regional bloc affected the probability of poor SRH.

Table 3 presents the results of the regression models used to predict the SRH scores in the three models (Models 1–3) using three variables: the number of new infection cases at the prefecture level (A), the average number of cases among neighboring prefectures (B), and their difference (B–A). As key explanatory variables, we used (A) in Model 1, (A) and (B) in Model 2, and (B) and (B–A) in Model 3. Model 3 was analyzed, which is statistically equivalent to Model 2, due to the linear relationship between (A), (B), and (B–A), to help interpret the results in Model 2. All models were adjusted for fixed effects at the individual and prefecture levels, as well as covariates at the individual level.

As presented in this table, an increase of 1000 cases at the prefecture level (A) increased the SRH score by 0.060 (95% confidence interval (CI): 0.044–0.077) in Model 1. In Model 2, however, (A) was considered non-significant, and the average score among neighboring prefectures (B) increased by 0.110 (95% CI: 0.055–0.166) per 1000 cases. This impact was equivalent to approximately one-tenth of the standard deviation of the SRH score. In Model 3, (B) increased the score by 0.113 (95% CI: 0.082–0.143), whereas the impact of (B–A) was negligible. This result confirms that only the average score among neighboring prefectures had a significant impact on the SRH; after controlling for it, the relative performance of the concerned prefecture did not affect SRH.

Table 4 summarizes and compares the results of the regression models to predict the SRH score (top part) and probability of poor SRH (bottom part) based on the variables of infection cases at different area levels (including the results reported in Table 3). Using only the cases in the prefecture as a regressor, an increase of 1000 cases modestly raised the SRH score by 0.60 (95% CI: 0.044–0.077), as already reported in Table 3, and the probability of poor SRH by 0.009 (95% CI: 0.002–0.016). Using the average cases among neighboring prefectures or in the regional bloc as a regressor together with the prefecture-level cases, the prefecture-level cases became non-significant, and the average at wider area levels was closely associated with the SRH score (columns 2 and 3). As in the case of the average among neighboring prefectures, an increase of 1000 cases in the regional bloc worsened the SRH score by 0.109 (95% CI: 0.065–0.153). By contrast, the prefecture-level cases had little impact on the SRH score.

A similar pattern was observed in the estimation results when the average was replaced with the maximum number of cases among neighboring prefectures or the maximum number of cases in the regional bloc; that is, only the maximum number of cases showed a significant impact on the SRH score (columns 4 and 5). However, the magnitude of the impact of the maximum number of cases—0.042 (95% CI: 0.024–0.061) among neighboring prefectures and 0.035 (95% CI: 0.021–0.048) in the regional bloc—was somewhat smaller than that of the average number of cases. The bottom part of the table provides the results for poor SRH, which again reveals the dominant importance of cases at the area levels wider than the prefecture. Unlike the SRH score, however, the average or the maximum number of cases among neighboring prefectures did not improve the SRH. We confirmed (but not reported in this table to save space) that the relative performance of the prefecture where individuals lived did not affect the SRH after controlling for the cases at wider areal levels, as already reported in Table 3.

Table 5 shows the estimation results of the impact of the declaration of the state of pandemic-related emergency, replacing the new infection cases with binary variables of the declaration at different area levels in the regression models to predict the SRH score and probability of poor SRH. Consistent with the results in Table 3, both SRH scores and poor SRH were affected only by the declaration of the state of emergency at wider area levels than that in a certain prefecture. For instance, the state of emergency declared within neighboring prefectures increased the SRH score by 0.227 (95% CI: 0.160–0.294) and increased the probability of poor SRH by 0.041 (95% CI: 0.013–0.070) (column 7). The state of emergency declared in the regional bloc has a similar impact (column 8).

## 4. Discussion

The association between an individual’s SRH and the number of new COVID-19 infection cases at different area levels was evaluated using longitudinal data obtained from a two-wave internet nationwide survey conducted during the pandemic in Japan. The reported number of new COVID-19 infection cases had an adverse impact on SRH, which is generally in line with the observations in previous studies that found an adverse impact of the pandemic on health outcomes [1,2,3,4,5]. In this sense, the reported number of new COVID-19 infection cases was a proxy of the perceived threat of COVID-19, and its harmful effect on SRH was consistent with the findings of previous studies [1,2,3,4,5]. The available information about COVID-19 infection may affect not only individuals’ concerns about risks of infection, restricted mobility, and economic hardship [6,7,8,10,11] but also perceived social isolation/loneliness and future anxiety [12,13,14,15,16,17,18,19,20,21,22,23]—all of which are likely to have adverse impacts on SRH.

A more noticeable finding was that SRH, both in terms of SRH score and the probability of poor SRH, was significantly affected by the information about infection in wider area levels than the information in a prefecture where the participant was living. The number of new COVID-19 infection cases and state of pandemic-related emergency in each prefecture have been reported every day in the mass media, based on the MHLW’s official press release. This process has allowed the individuals to know the situations not only in their prefecture but also in their neighboring prefectures and the regional bloc. After jointly using the infection cases at the prefecture level and the average or maximum number of cases in neighboring prefectures or in the regional bloc, the results showed that SRH was only affected by infection cases at wider area levels. Consistently, SRH was more affected by the declaration of the state of emergency in areas wider than the prefecture.

These results underscore that individuals were concerned about the number of infection cases in a wider area compared with that in a prefecture. Even if an individual is living in a prefecture with a low infection rate, the high infection rate in neighboring prefectures is important for making subjective health assessments. This is a reasonable result, given the high risk of infection in a neighboring prefecture, but it is not supportive of the view that comparisons with neighboring prefectures matter, contrary to the “relative income hypothesis” regarding subjective well-being [24,25,26]. However, it is somewhat surprising that individuals did not respond to infection cases at the prefecture level after controlling for the impact of cases at wider area levels. This was also true for the declaration of the state of emergency. These results suggest that individuals tend to evaluate the risk of COVID-19 in a relatively wide geographic framework. Moreover, the magnitude of the impact of the average number of cases among neighboring prefectures or in the regional bloc was somewhat larger than that of the maximum number of cases. Individuals may consider that the average number of cases represent the area-level risk of infection more accurately than the maximum number of cases, probably because the latter were considered to overstate the risk.

This study has several limitations in addition to the limited sample size, limited national representativeness of the sample, and potential biases inherent in an internet survey. First, SRH, which was used to represent the respondent’s overall conditions, was subjectively assessed and thus not free from measurement errors. Second, the mechanism linking the reported infection cases to SRH was not examined, even though individuals can obtain updated information about the infection through the mass media every day. The impact of such information on mental health and health behavior should be explored further to identify the underlying mechanisms. Third, the dynamic impact of reported infection cases was not considered, which may have confounded the association between area-level infection cases and SRH. Individuals are likely to adopt to the reported high levels of cases over time and are also likely to respond positively to even a small reduction in the number of cases. Fourth, caution should be observed when generalizing the results of this study because policy measures to restrict mobility and contain infection vary substantially by country. These issues should be addressed in future research.

## 5. Conclusions

This study showed that an individual’s SRH was affected by the number of infection cases in a wider area compared with that in the prefecture. Results suggest that inter-prefectural coordination is needed not only to contain COVID-19 but also to reduce its adverse impact on the subjective health outcomes of residents.

## Figures and Tables

**Figure 1 ijerph-19-08918-f001:**
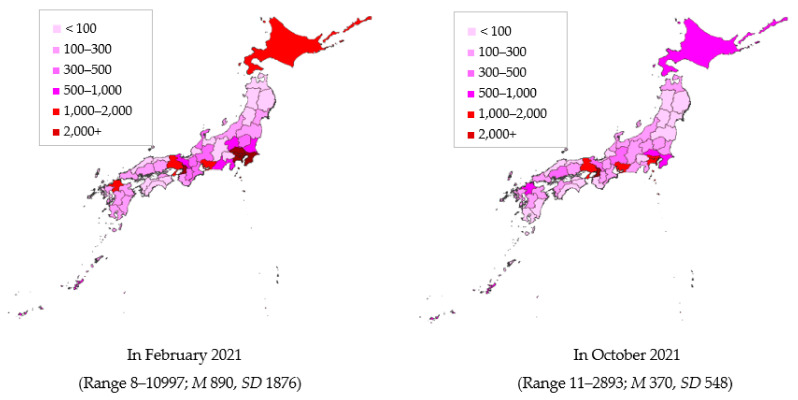
New COVID-19 infection cases by prefecture.

**Table 1 ijerph-19-08918-t001:** Key features of participants evaluated in Wave 1.

a		All		Men	Women
Married		56.3		52.7	56.6
Educational attainment					
Junior high school		2.6		2.9	2.3
High school		42.4		39.0	45.4
Junior college		12.9		4.5	20.3
College or above		42.1		53.6	32.0
Occupational status					
Regularly employed		43.7		60.3	29.1
Non-regularly employed		21.7		12.6	29.7
Self-employed		6.7		8.9	4.7
Unemployed		2.6		1.7	3.3
Out of labor force		21.8		12.6	29.8
Students		3.6		3.8	3.4
Regional bloc ^a^					
Hokkaido		2.9	(4.1)	2.8	3.0
Tohoku		10.9	(6.8)	11.5	10.4
Kanto		21.1	(34.7)	21.6	20.6
Chubu		19.2	(18.1)	18.7	19.7
Kinki		14.6	(16.3)	14.7	14.4
Chugoku/Shikoku		16.4	(8.7)	16.0	16.7
Kyushu		12.7	(10.1)	12.5	12.8
Okinawa		2.2	(1.2)	2.1	2.2
Age	*M*	44.1		43.6	44.5
(years)	*SD*	15.3		15.3	15.3
Household income	*M*	601.5		677.4	535.3
(annual, million JPY)	*SD*	717.1		898.5	499.3
*N*		2260		1054	1206

^a^ Numbers in parentheses indicate the actual proportions of population in 2021.

**Table 2 ijerph-19-08918-t002:** Pairwise correlation analysis between self-rated health and COVID-19 infection (*N* = 4520 observation of 2260 individuals).

	a	SRH Score		Poor SRH	
Range		1 (Good) to 5 (Poor)		0–1	
*M*		2.61		0.20	
*SD*		(1.07)		(0.40)	
Pairwise correlation with:					
New infection cases					
	Prefecture	0.010		0.010	
	Neighboring prefecture average	0.036	*	0.015	
	Regional bloc average	0.055	***	0.030	*
	Neighboring prefecture max	0.038	*	0.014	
	Region max	0.053	***	0.029	*
State of emergency					
	Prefecture	0.032	*	0.006	
	Neighboring prefectures	0.085	***	0.024	
	Regional bloc	0.085	***	0.037	*

** p* < 0.05, *** p* < 0.01, **** p* < 0.001.

**Table 3 ijerph-19-08918-t003:** Estimated association between new COVID-19 cases and self-rated health (SRH) score ^a^ (*N* = 4520 observations of 2260 individuals).

Dependent Variable = SRH Score: 1 (Good) to 5 (Poor)	Model 1	Model 2	Model 3
New infection cases (×1000)			^a^
Prefecture (A)	0.060	0.002	
	(0.044, 0.077)	(–0.032, 0.036)	
Neighboring prefectures average (B)		0.110	0.113
		(0.055, 0.166)	(0.082, 0.143)
Difference (B–A)			0.002
			(–0.032, 0.036)

^a^ Controlled for fixed effects at individual and prefecture levels as well as covariates at the individual level. A set of full estimation results is available upon request from the authors. Numbers in parentheses indicate 95% confidence interval.

**Table 4 ijerph-19-08918-t004:** Estimated association of new COVID-19 cases with self-rated health (SRH) ^a^ (*N* = 4520 observations of 2260 individuals).

	(1)	(2)	(3)	(4)	(5)
Dependent variable = SRH score: 1 (good) to 5 (poor)					** ^a^ **
New infection cases (×1000)					
Prefecture	0.060	0.002	0.013	0.003	0.013
	(0044, 0.077)	(–0.032, 0.036)	(–0.012, 0.039)	(–0.027, 0.034)	(–0.012, 0.038)
Neighboring prefectures average		0.110			
		(0.055, 0.166)			
Regional bloc average			0.109		
			(0.065, 0.153)		
Neighboring prefectures maximum				0.042	
				(0.024, 0.061)	
Regional bloc maximum					0.035
					(0.021, 0.048)
Dependent variable = Poor SRH					
New infection cases (× 1000)					
Prefecture	0.009	0.004	–0.001	0.003	–0.001
	(0.002, 0.016)	(–0.010, 0.019)	(–0.011, 0.010)	(–0.010, 0.015)	(–0.011, 0.010)
Neighboring prefectures average		0.009			
		(–0.014, 0.032)			
Regional bloc average			0.022		
			(0.004, 0.040)		
Neighboring prefectures maximum				0.005	
				(–0.003, 0.012)	
Regional bloc maximum					0.007
					(0.001, 0.013)

^a^ Controlled for fixed effects at the individual and prefecture levels as well as covariates at the individual level; full estimation results are available upon request from the authors.

**Table 5 ijerph-19-08918-t005:** Estimated association between the state of emergency and self-rated health (SRH) ^a^ (*N* = 4520 observations of 2260 individuals).

	(6)	(7)	(8)
Dependent variable = SRH score: 1 (good) to 5 (poor)			^a^
State of emergency			
Prefecture	0.279	0.056	0.045
	(0.214, 0.345)	(–0.037, 0.150)	(–0.039, 0.129)
Neighboring prefectures		0.227	
		(0.160, 0.294)	
Regional bloc			0.240
			(0.187, 0.294)
Dependent variable = Poor SRH (binary)			
State of emergency			
Prefecture	0.033	–0.008	–0.001
	(0.006, 0.060)	(–0.047, 0.031)	(–0.036, 0.033)
Neighboring prefectures		0.041	
		(0.013, 0.070)	
Regional bloc			0.035
			(0.013, 0.057)

^a^ Controlled for fixed effects at the individual and prefecture levels as well as covariates at the individual level; full estimation results are available upon request from the authors.

## Data Availability

Data supporting reported results are available from https://www5.cao.go.jp/keizai2/wellbeing/manzoku/index.html.

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
