# Peer review of "At Which Area Level Does COVID-19 Infection Matter Most for an Individual’s Self-Rated Health? A Multilevel Fixed-Effects Model Analysis in Japan"

_ijerph, 2022, doi:10.3390/ijerph19158918_

Round 1

Reviewer 1 Report

The authors had better tools for assessing health status, as perceived by each individual.

Although the methodology and statistical analysis seem to be consistent with the objectives, the usefulness of the results I think is affected by the simplicity of the evaluation.

Author Response

We sincerely appreciate your comments and suggestions, which we find are all constructive and helpful. In what follows, we explain how we responded to them. The line numbers in the parentheses are in the revised manuscript (with tracked changes).

The authors had better tools for assessing health status, as perceived by each individual.

=> We recognize both the merit and limitation of self-rate health (SRH). We added the explanation about the merit of self-rated health: “SRH is frequently used in demographic and population health surveys to capture respondents’ self-reported general health including mortality [28,29]. The merit of this self-reported indicator is explained by its ease of use (a single question) and its validity and reliability [28,30,31]” (Lines 120-123). We also added the limitation in Discussion (see below).

Although the methodology and statistical analysis seem to be consistent with the objectives, the usefulness of the results I think is affected by the simplicity of the evaluation.

=> We added your point as the first limitation in Discussion: “First, SRH, which was used to represent the respondent’s overall conditions, was subjectively assessed and thus not free from measurement errors” (Lines 330-331).

Reviewer 2 Report

The manuscript addresses the relevant topic of self rated health following COVID-19 infection. The manuscript is overall well-written and data clearly presented.

The authors are suggested:

- To specify if the questionnaire was pre-tested on a sample population to evaluate the presence of potential issues in questions' interpretation

- To add sample size estimation

- To add the questionnaire as supplementary material/appendix

Author Response

We sincerely appreciate your comments and suggestions, which we find are all constructive and helpful. In what follows, we explain how we responded to them. The line numbers in the parentheses are in the revised manuscript (with tracked changes).

The manuscript addresses the relevant topic of self rated health following COVID-19 infection. The manuscript is overall well-written and data clearly presented.

The authors are suggested:

- To specify if the questionnaire was pre-tested on a sample population to evaluate the presence of potential issues in questions' interpretation

=> We actually pretested the questionnaire, and added: “Before the survey was conducted, the questionnaire was pretested to explore potential issues in interpreting the questions” (Lines 94-96).

- To add sample size estimation

=> We added the results of sample size estimation: “By setting the effect size = .35, α = .05, the power (1- β) = .8, and the number of predictors = 2,324 (which reflected 19 independent variables, 2,260 individual fixed effects, and 47 prefectural fixed effects), the sample size was estimated to be 2,462, which was well below 4,520 observations used in our regression analyses” (Lines 1769-172).

- To add the questionnaire as supplementary material/appendix

=> We added the full set of the questionnaire as the supplementary file and mentioned it in Lines 104 and 349-350.

Reviewer 3 Report

Dear Authors please find my comments below:

1. Introduction:

line 31 - please correct the following ..."of lockdowns and social distancing plolicies"

2. Methodoly - if the patients were asked whether they had been infected with COVID, or at the moment were ill, or someone from their family suffers orm died? This is very important question because the disese within the family or of a person migth strongly impact the self health assessment. Please clarify here and also discuss in the discussion section.

3. Discussion: What the mass media reported? If they reported the prefecture cases and not the neighboring area, or wirder area, or nearby block the people will be affected logically by that messages. 

The authors stated as a limitation that they do not explore the mechanism linking the reported infection cases to SRH but they should at least comment on the knowledge for the disease cases. From where the people understand the information about the disease and which area it covers. 

In the discussion section the authors must comment on the orgnaisation of process of informing people in the country and within the explored habitats. 

The authors must give more insigth about the connection between the available information for COVID cases and SRH.   

Author Response

We sincerely appreciate your comments and suggestions, which we find are all constructive and helpful. In what follows, we explain how we responded to them. The line numbers in the parentheses are in the revised manuscript (with tracked changes).

  1. Introduction:

line 31 - please correct the following ..."of lockdowns and social distancing plolicies"

=> Thank you for pointing out the typo. We corrected it.

  1. Methodoly - if the patients were asked whether they had been infected with COVID, or at the moment were ill, or someone from their family suffers orm died? This is very important question because the disese within the family or of a person migth strongly impact the self health assessment. Please clarify here and also discuss in the discussion section.

=> In response to your constructive comments, we added the experiences of (i) COVID-19 infection and (ii) other serious disease or injury over the past one year prior to the survey time as explanatory variables (see Lines 148-150). Correspondingly, we redid all regression analyses and revised the figures in Tables 3­-5 and in the text, and found that the estimation results remained virtually unchanged, leaving the key message generally intact. However, please note that the survey did not asked about the COVID-related health conditions of family members.

  1. Discussion: What the mass media reported? If they reported the prefecture cases and not the neighboring area, or wirder area, or nearby block the people will be affected logically by that messages. 

=> In response to your comments, we added: “The number of new COVID-19 infection cases and state of pandemic-related emergency in each prefecture have been reported every day in the mass media, based on the MHLW’s official press release. This process has allowed the individuals to know the situations not only in their prefecture but also in their neighboring prefectures and the regional bloc” (Lines 303-307). Note that MHLW is Ministry of Health, Labour and Welfare (spelled out in Line 135).

The authors stated as a limitation that they do not explore the mechanism linking the reported infection cases to SRH but they should at least comment on the knowledge for the disease cases. From where the people understand the information about the disease and which area it covers. 

=> After stating: “the mechanism linking the reported infection cases to SRH was not examined,” we added: “even though the individuals can obtain updated information about the infection through the mass media every day” (Lines 332-334).

In the discussion section the authors must comment on the orgnaisation of process of informing people in the country and within the explored habitats. 

=> As mentioned above, we added: “The number of new COVID-19 infection cases and state of pandemic-related emergency in each prefecture have been reported every day in the mass media, based on the MHLW’s official press release. This process has allowed the individuals to know the situations not only in their prefecture but also in their neighboring prefectures and the regional bloc” (Lines 303-307).

The authors must give more insigth about the connection between the available information for COVID cases and SRH.   

=> We added: “The available information about COVID-19 infection may affect not only individuals’ concerns about risks of infection, restricted mobility, and economic hardship [6–8,10,11] but also perceived social isolation/loneliness and future anxiety [12–23] – all of which are likely to have adverse impact on SRH” (Lines 296-300).